# Quaternion Valued Risk Diversification

**DOI:** 10.3390/e22040390

**Published:** 2020-03-29

**Authors:** Seisuke Sugitomo, Keiichi Maeta

**Affiliations:** 1Fund Manager at Epic Partners Investment Co., Ltd., Tokyo 100-0013, Japan; 2Graduate School of Mathematical Sciences, University of Tokyo, Tokyo 113-8654, Japan

**Keywords:** portfolio management, risk diversification, Hilbert transform, principal component analysis, quaternion

## Abstract

Risk diversification is an important topic for portfolio managers. Various portfolio optimization algorithms have been developed to minimize portfolio risk under certain constraints. As an extension of the complex risk diversification portfolio proposed by Uchiyama, Kadoya, and Nakagawa in January 2019 (Yusuke et al. *Entropy.*
**2019**, *21*, 119.), we propose a risk diversification portfolio construction method which incorporates quaternion risk. We show that the proposed method outperforms the conventional complex risk diversification portfolio method.

## 1. Introduction

A combination of multiple financial assets generated as a result of deciding which and how much to invest in multiple financial assets is called a portfolio. When constructing a portfolio, a method for quantitatively determining the risk using a mathematical method and minimizing the risk under certain constraints is called a portfolio optimization method. Many portfolio optimization algorithms have been devised that can mathematically minimize portfolio risk. The most famous portfolio optimization algorithm is the mean variance (MV) approach proposed by Markowitz [1]. The idea is to reduce risk by combining different assets with low correlation, but it has been pointed out that the weight of a specific asset may become too high [2].

A risk parity (RP) portfolio is a portfolio which is constructed to equalize the contribution of each asset to the overall portfolio risk [3]. On the other hand, it has been pointed out that risk parity portfolios have a uniform risk contribution in each asset, but this does not necessarily diversify the sources of the risk.

Meucci [4] proposed a new measure of portfolio risk variance called the variance index. Maximum risk diversification (MRD) portfolios have been proposed, which use principal component analysis to construct a principal component portfolio which is uncorrelated from the original assets and equalizes the risk contribution of each principal component portfolio to the portfolio risk. MRD portfolios have also been confirmed to outperform MV portfolios and to be able to allocate the risk contribution of assets [5]. After the concept and method for the MRD portfolio construction were proposed, the method was tested and has been used both in academia and industry. Recently, in fact, the availability of MRD portfolio construction has been reported in an empirical test for commodities [6].

The conventional portfolio construction methods described above are constructed based on the covariance matrix estimated from the time-series of the return of the assets. However, this covariance matrix cannot incorporate dynamic information of the time-series, in terms of the return of the assets. Therefore, Uchiyama, Kadoya, and Nakagawa proposed a method called complex valued risk diversification (CVRD) [7]. The traditional risk diversification portfolio was inspired by empirical orthogonal function methods developed in the field of meteorological physics. In the empirical orthogonal function method, the complex orthogonal function is determined by performing principal component analysis on the Hermitian matrix of the analytic signal, as an evolution of the method that uses principal component analysis of the Hermitian matrix of the time-series data (corresponding to the covariance matrix of the returns in the risk diversification portfolio). As the complex orthogonal function includes phase information in addition to amplitude, it is possible to extract dynamic information relating to a time delay inherent in the target time-series data. After forming the complex variable time-series from Hilbert-transformed return series of each asset, calculating the complex valued matrix, and constructing the risk diversification portfolio, the authors realized risk diversification including dynamic risk information while outperforming the traditional portfolio construction methods.

In this paper, we propose the quaternion valued risk diversification (QVRD) method, which extends CVRD by naturally extending the complex valued covariance matrix to a quaternion valued covariance matrix. We use a quaternion signal, which has a four-dimensional information of amplitude and three phases, in order to try to more precisely find signal movements which cannot be observed in a complex signal.

## 2. Conventional Methods

In this section, we review conventional portfolio construction methods. We use the following notation:M∈N: the number of assets${\left\{p_{t}^{\left(m\right)}\right\}}_{0<t<T}$: $m^{\mathrm{th}}$ asset prices$r_{t}^{\left(m\right)}=\frac{p_{t+1}^{\left(m\right)}-p_{t}^{\left(m\right)}}{p_{t}^{\left(m\right)}}$: the return of $m^{\mathrm{th}}$ asset prices$\Sigma =E[\left(r_{t}-E\left[r_{t}\right]\right){\left(r_{t}-E\left[r_{t}\right]\right)}^{T}]$: the covariance matrix$w=(w_{m})$: a portfolio (vector)$R_{t}=\sum_{m=1}^{M}w_{m}r_{t}^{m}$: the return of the portfolio *w*$\sigma^{2}=w^{T}\Sigma w$: the variance of the portfolio *w*

### 2.1. Mean Variance Portfolio

Markowitz first introduced MV portfolio construction as a sophisticated method in modern portfolio theory. In this theory, the risk of an asset is defined as the standard deviation of the return. Given μ∈R and $\left\{p_{m}\right\}$, the MV optimization with expected return μ can be described as follows:
**Problem** **1.**Find the portfolio w which minimizes $\sigma^{2}$ subject to $R_{t}=\mu$.

We can solve this problem using the Lagrange multipliers method, under some constraint with respect to the portfolio *w*.

### 2.2. Risk Parity Portfolio

It has been pointed out that the asset class of a MV portfolio may not be fully allocated. To disperse the risk contributions of portfolios, RP portfolio construction has been proposed. The key idea of an RP portfolio is equalizing the risk of contributions.

The risk contributions of the $m^{\mathrm{th}}$ asset are derived from the variance as follows:$$\sigma_{m}=w_{m}\frac{\partial \sigma }{\partial w_{m}}=w_{m}\frac{{\left(\Sigma w\right)}_{m}}{\sigma }.$$

Equalizing the risk contributions (minimize $\left(\sigma_{m}-\frac{\sigma }{M}\right)$), we have the RP optimization:

**Problem** **2.**
*Find the portfolio w which minimizes $\sum_{m=1}^{M}\left(w_{m}-\frac{\sigma^{2}}{{\left(\Sigma w\right)}_{m}M}\right)$.*


### 2.3. Maximum Risk Diversification Portfolio

In general, the origins of the risks of assets seem to be entangled; namely, the covariance matrix of the return of a portfolio contains non-diagonal components. To unravel entangled risks, principal component analysis has been incorporated into portfolio constructions [4].

As the covariance matrix Σ is symmetric, it can be transformed into a diagonal matrix by using an appropriate orthogonal matrix:$$\Lambda =U^{T}\Sigma U=\mathrm{diag}\left\{\lambda_{m}\right\}.$$

The eigenvalues $\left\{\lambda_{m}\right\}$ introduce a probability distribution of risk contribution. Thus, the entropy *H* with respect to the probability distribution is defined and is employed as the objective function of the MRD portfolio construction:
**Problem** **3.**Find the portfolio w which maximizes the entropy H.

As is well-known, the principle of maximum entropy introduces the most diversified probability distribution under given constraints. In this case, such a probability distribution of risk contributions is desirable for allocating the origin of risks related to the assets of a portfolio.

## 3. Complex Valued Risk Diversification Portfolio Construction

In this section, we review the CVRD method, according to Uchiyama, Kadoya, and Nakagawa [7], which is the conventional method used for comparison in this paper.

### 3.1. Complex Hilbert Transform and Analytic Signal

**Definition** **4**(Complex Hilbert transform). *The Hilbert transform of a real signal x(t) on t∈[0,∞) is defined by:*
(1)$$H\left[x\left(t\right)\right]=\frac{1}{\pi }\int_{0}^{\infty }\frac{x(\tau )}{t-\tau }d\tau ,$$
*where the improper integral is understood in the sense of Cauchy’s principal value [8]. In practice, empirical time-series are recorded at a certain sampling rate Δt, which introduces a discrete time $t_{n}=n\Delta t$, with n being an integer. The Hilbert transform for a discrete time-series is given by:*
(2)$$H_{D}\left[x_{k}\right]=-i\mathrm{sgn}\left(k-\frac{N}{2}\right)\sum_{n=0}^{N-1}x_{n}e^{i\frac{2\pi n}{N}},$$

*where sgn(·) is the sign function [9].*


**Definition** **5**(Analytic signal). *For a real signal x(t), we say $x\left(t\right)+iH\left[x_{t}\right]$ is the analytic signal of x(t).*

### 3.2. Procedure

In this section, we review the procedure of CVRD using the following notation:M∈N: the number of assets${\left\{p_{t}^{\left(m\right)}\right\}}_{0<t<T}$: $m^{\mathrm{th}}$ asset prices$r_{t}^{\left(m\right)}=\frac{p_{t+1}^{\left(m\right)}-p_{t}^{\left(m\right)}}{p_{t}^{\left(m\right)}}$: the return of $m^{\mathrm{th}}$ asset prices$w\in \Delta^{n}:=\left\{\left(w_{m}\right)|w_{m}>0,\;w_{1}+\cdots +w_{m}=1\right\}$: a portfolio (vector)

We apply the Hilbert transform to the return of the portfolio and obtain the analytic signal as:(3)$$z_{t}=r_{t}+iH_{D}\left[r_{t}\right].$$

As well as the principal component analysis for real valued time-series, the analytic signal $z_{t}$ provides a complex valued covariance matrix, defined as:(4)$$C_{z}=E\left[\left(z_{t}-E\left[z_{t}\right]\right){\left(z_{t}-E\left[z_{t}\right]\right)}^{*}\right],$$
where $z_{t}^{*}:={\bar{z_{t}}}^{T}$. As $C_{z}$ is a Hermitian matrix, it is diagonalizable with unitary matrix *U*. Note that $C_{z}$ is a positive definite Hermitian matrix. The eigenvalues of $C_{z}$ are positive real values (including zeros), and the unitary matrix *U* transforms $C_{z}$ into:(5)$$U^{*}C_{z}U=\Lambda ,$$
where Λ is an orthogonal matrix which consists of the eigenvalues $\lambda_{n}$ of $C_{z}$. The portfolio risk can be estimated as $v_{m}:={\left|{\tilde{w}}_{m}\right|}^{2}\lambda_{m}$ under $\tilde{w}=U^{*}w$, with ${\tilde{w}}_{m}$ being the $m^{\mathrm{th}}$ component of the transformed weight coefficient $\tilde{w}$, and the probability distribution for $v_{m}$ is defined by:(6)$$p_{m}=\frac{v_{m}}{\sum_{m=1}^{M}v_{m}}.$$

From this probability distribution, the corresponding entropy can be introduced as:(7)$$H=-\sum_{m=1}^{M}p_{m}\log p_{m}.$$

In general, the weight coefficients of portfolios are constrained based on trading strategies. Thus, we constructed a Lagrangian function with the entropy *H* and constraint functions $g_{l}$ for weight coefficients as:(8)$$L=H-\sum_{l=1}^{L}\mu_{l}g_{l}\left(\tilde{w}\right),$$
where $\mu_{l}$ is a Lagrange multiplier. Optimizing *L* with respect to $w\in R^{n}$ gives the weight coefficients of the expected CVRD portfolio.

## 4. Quaternion Valued Risk Diversification Portfolio Construction

In this section, we review quaternions [10] and discuss the procedure for building a quaternion valued risk diversification portfolio.

### 4.1. Quaternion

**Definition** **6**(Quaternion, Conjugate). *We define the R−algebra H by a four-dimensional real vector space:*
$$H=\left\{x_{0}+x_{1}i+x_{2}j+x_{3}k\;:\;x_{0},x_{1},x_{2},x_{3}\in R\right\},$$
*with the following multiplication operations:*
$$i^{2}=j^{2}=k^{2}=ijk=-1.$$
*An element of H is called a quaternion. For a quaternion $x=x_{0}+x_{1}i+x_{2}j+x_{3}k$, the quaternion $\bar{x}=x_{0}-x_{1}i-x_{2}j-x_{3}k$ is called the conjugate of x.*


**Remark** **7.**
*We note some basic remarks:*

*$H=R⊕Ri⊕Rj⊕Rk≃C⊕Cj≃R^{4}$.*

*Multiplication of two quaternions is non-commutative (e.g., ij=−ji).*



**Definition** **8**(Adjoint, Eigenvalue, Hermitian matrix). *Let n and m be natural numbers. We denote by M(n,m,H) the set of n×m matrices and by M(n,H) the set of n-dimensional square matrices with quaternion coefficients:*
For a matrix $X=\left\{x_{ij}\right\}\in M\left(m,n,H\right)$, the matrix $X^{*}:={\bar{X}}^{T}=\left\{{\bar{x}}_{ji}\right\}\in M\left(n,m,H\right)$ is called the adjoint of X.For a matrix X∈M(n,H), we say λ∈H is a (right) eigenvalue and $v\in H^{n}\backslash \left\{0\right\}$ is an eigenvector of X if Xv=vλ.A matrix X∈M(n,H) is called a Hermitian matrix if $X=X^{*}$.

**Fact** **9**([10]). *A Hermitian matrix is diagonalizable and its eigenvalues are real.*

### 4.2. Quaternion Fourier Transform and Quaternion Signal

In this section, we review the quaternion Fourier transform, according to Bihan, Sangwine, and Aug [11].

**Definition** **10.**
*For a complex valued function z(t), the quaternion Fourier transform is defined as:*
$$F_{j}\left[z\right]\left(\nu \right)=\int_{-\infty }^{\infty }z\left(t\right)e^{-j2\pi \nu t}dt.$$

*The inverse transform is defined as:*
$$F_{j}^{-1}\left[z\right]\left(t\right)=\int_{-\infty }^{\infty }z\left(\nu \right)e^{j2\pi \nu t}d\nu .$$


As well as the complex case, a quaternion Hilbert transform can be defined, as follows:

**Definition** **11.**
*With z(t) being a complex signal, the quaternion Hilbert transform $H_{j}\left[z\left(t\right)\right]$ is defined as:*
$$H_{j}\left[z\left(t\right)\right]=F_{j}^{-1}\left[-j\mathrm{sgn}\left(\nu \right)F_{j}\left[z\right]\right]\left(\nu \right).$$

*The quaternion signal (see [9] (hypercomplex signal representation)) is defined as:*
$$\hat{z}\left(t\right)=z\left(t\right)+jH_{j}\left[z\left(t\right)\right]$$


### 4.3. Procedure

With the preparations so far, we can see that the procedure 2.2 can be appropriately generalized to quaternions. Specifically, the procedure can be changed as follows:After derivation of the analytic signal (3), transform it to the quaternion signal:
$${\hat{z}}_{t}=z\left(t\right)+jH_{j}\left[z\left(t\right)\right].$$Instead of the complex valued covariance matrix (4), we use the quaternion valued covariance matrix, defined as:
$$Q_{\hat{z}}=E\left[\left({\hat{z}}_{t}-E\left[{\hat{z}}_{t}\right]\right){\left({\hat{z}}_{t}-E\left[{\hat{z}}_{t}\right]\right)}^{*}\right].$$Note that $Q_{\hat{z}}$ is a positive definite Hermitian matrix.By Fact 9, the matrix $Q_{\hat{z}}$ is diagonalizable and its eigenvalues are real; namely,
$$U^{*}Q_{\hat{z}}U=\Lambda .$$Use this instead of Equation (5).

## 5. Empirical Analysis

In this section, we test the performance of the proposed quaternion valued risk diversification portfolio by comparing it with the complex valued risk diversification portfolio method.

### 5.1. Data Description

To evaluate the performance of each portfolio, we selected four commodities, six indices, and five global currencies (to US dollar), as shown in Table 1. All daily historical data were collected during January 2005 to November 2019 and transformed into returns to estimate the covariance matrices. The descriptive statistics of the returns of the assets are shown in Table 2.

### 5.2. Performance Test of Portfolios

In order to implement the portfolio optimization methods, we used a complex valued empirical covariance matrix for CVRD portfolio construction and a quaternion valued empirical covariance matrix for QVRD portfolio construction. It has been pointed out, in the random matrix theory, that the empirical covariance matrix cannot infer the true value of a covariance matrix [12,13,14]. Nevertheless, the empirical covariance matrix has been used to construct portfolios in industry, due to ease of estimation for practitioners. Based on the correction formula proposed in [14], the empirical return of the portfolio is $1/\sqrt{1-p/n}$ times larger than its true value, where *p* is number of assets in the portfolio and *n* is the length of the historical data. In this case, *p* = 15 and *n* = 252 and, so, the correction ratio was evaluated as $1/\sqrt{1-p/n}\approx 1.031$. Thus, we believe that the empirical covariance matrix can be used to better infer the true covariance matrix.

All portfolios were rebalanced on a monthly basis, and the one year daily data from the analyzing month was used for each covariance matrix estimate. Transaction costs were considered as 10 bps per transaction. Cumulative returns, standard deviations, and Sharpe ratios were used as performance indicators. These are all commonly used indicators, in practice. Although Sharpe ratios are commonly used, the standard deviation, which was adopted in the Sharpe ratios, is not a good measure of risk as it penalizes upside deviation, as well as downside deviation. Farinelli and Tibiletti [15] proposed a FT ratio that exclusively looks at the upper and lower partial moments by comparing the favorable and unfavorable events. Formally, for any prospect *X*, its FT ratio is defined as:(9)$$\phi_{FT,X}\left(\eta \right)=\frac{{\left(E\left[{\left(X-\eta \right)}_{+}^{p}\right]\right)}^{1/p}}{{\left(E\left[{\left(\eta -X\right)}_{+}^{q}\right]\right)}^{1/q}},$$
where $x_{+}$ = max0, x and η is called the return threshold. For any investor, return below their return threshold is considered loss and return above is gain. Furthermore, *p* and *q* are positive values which represent an investor’s degree of risk aversion. Thus, the FT ratio is the ratio of average gain to average loss, each raised by some power index to proxy for the investor’s degree of risk aversion. We used 0 for the threshold return and two different combinations of *p* and *q*.

For moderate investors, we used *p* = *q* = 1, which is called the Omega ratio. For conservative investors, we used *p* = 0.5 and *q* = 2 [16,17].

Table 3 shows the cumulative return, monthly return standard deviation, Sharpe ratio, omega ratio, and FT ratio for each portfolio. The QVRD portfolio outperformed the RP and the MRD and the CVRD portfolio in terms of return, Sharpe ratio, omega ratio, and FT ratio.

Table 4 shows the average weight of each asset for each portfolio. The major difference between the QVRD portfolio and the CVRD portfolio is the weight on the currency, which was basically a lower weight for the QVRD.

Figure 1, Figure 2, Figure 3, Figure 4, Figure 5 and Figure 6 show the weight sequences of each asset corresponding to the RP, CVRD, and QVRD portfolios for each term. It can be seen that the weight changes were more gradual in the QVRD portfolio than in the CVRD portfolio. Therefore, it was considered that the rebalancing cost can be further reduced.

Figure 7, Figure 8 and Figure 9 show the cumulative returns for all terms, from 2005 to 2010, and from 2011 to 2019, respectively, for each approach. The QVRD portfolio constantly outperformed the other approaches. In particular, the QVRD portfolio significantly outperformed each portfolio from 2017. It is likely that expanding to quaternions captured the more complex risk characteristics of the financial assets.

## 6. Conclusions

The complex risk diversification portfolio method, proposed by Uchiyama, Kadoya, and Nakagawa in January 2019 [7], was the first to include dynamic asset information in portfolio optimization. We tried to include more complex dynamic information in portfolio optimization by naturally extending complex risk to quaternion risk.

The conventional method, which uses the Hilbert transform, is equivalent to performing a two-dimensional amplitude and phase analysis by viewing the real signal as a projection of a complex signal. In this paper, we use a quaternion signal, which has four-dimensional information of amplitude and three phases. Just as we can detect more detailed signal movements by considering complex signals, we can find signal movements more precisely that cannot be observed in a complex signal. From the results, it can be said that it is better to use the quaternion signal, compared to the complex one, especially after 2013. It is considered that, as the global financsial markets became more complex after 2013, risk properties have become harder to detect.

As a result, the QVRD portfolio outperformed the CVRD portfolio in key performance indicators, such as cumulative return, Sharpe ratio, and omega ratio. In practice, changing the time period for estimating the quaternion valued covariance matrix or changing the limits of portfolio construction enables more optimal portfolios for investors.

Furthermore, we can consider the dynamic risks using a Clifford algebra, in principle, in the same way. In the future, we may extend the concept of the risk to Clifford algebras.

## Figures and Tables

**Figure 1 entropy-22-00390-f001:**
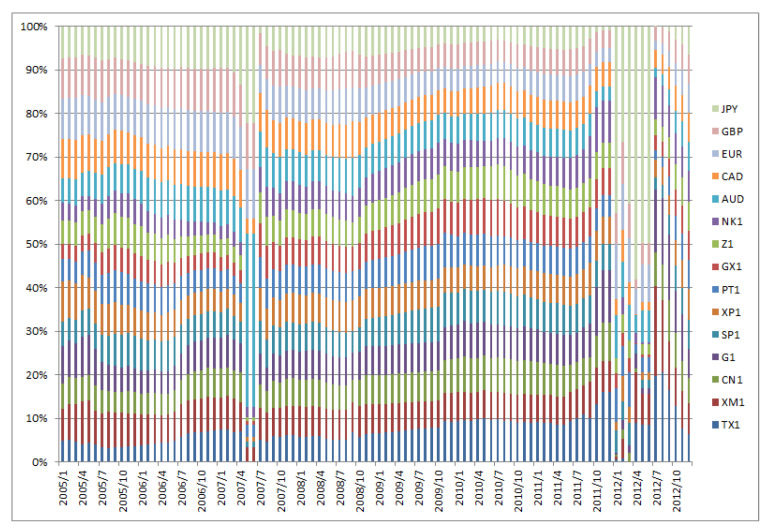
The allocation of the assets in the risk parity (RP) portfolio construction from 2005 to 2012.

**Figure 2 entropy-22-00390-f002:**
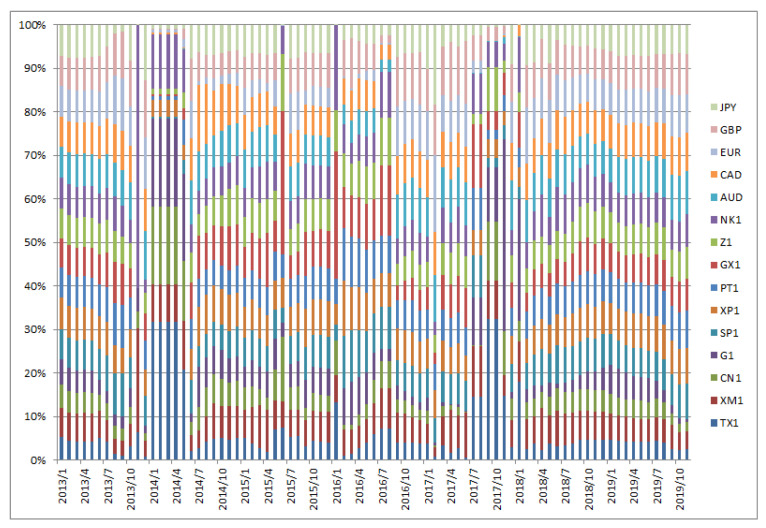
The allocation of the assets in the risk parity (RP) portfolio construction from 2013 to 2019.

**Figure 3 entropy-22-00390-f003:**
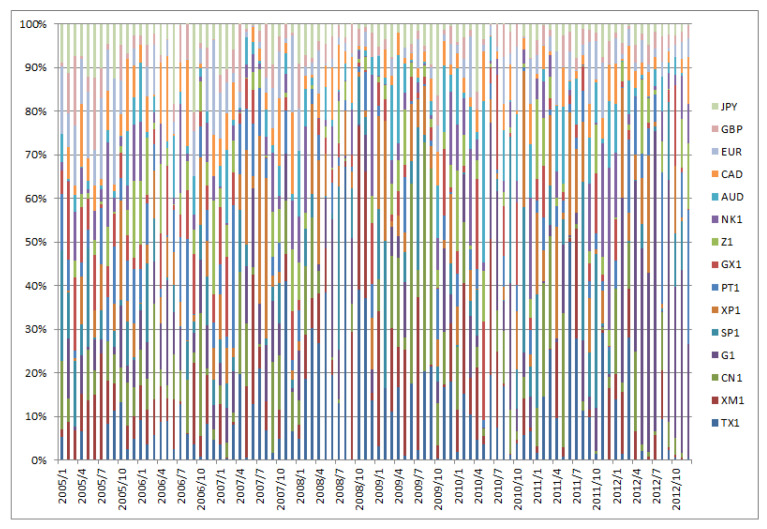
The allocation of the assets in the complex valued risk diversification (CVRD) portfolio construction from 2005 to 2012.

**Figure 4 entropy-22-00390-f004:**
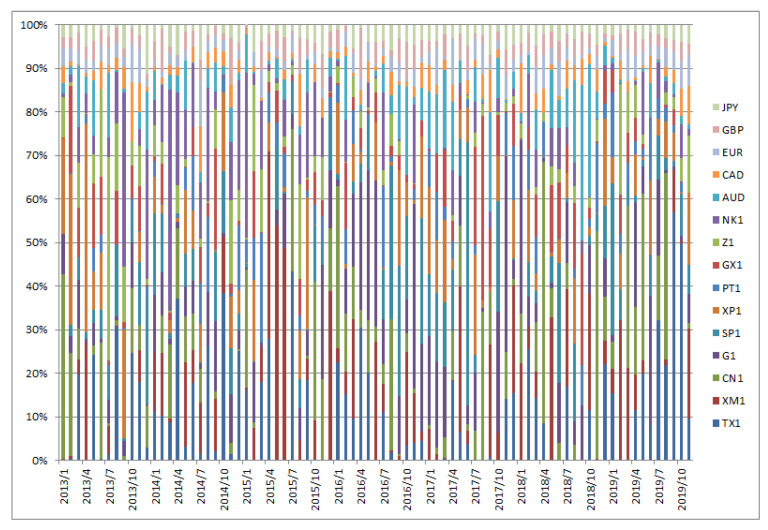
The allocation of the assets in the complex valued risk diversification (CVRD) portfolio construction from 2013 to 2019.

**Figure 5 entropy-22-00390-f005:**
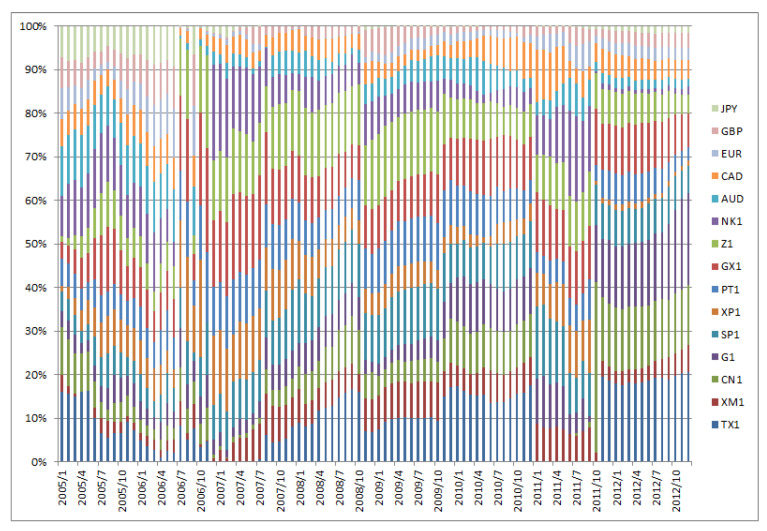
The allocation of the assets in the quaternion valued risk diversification (QVRD) portfolio construction from 2005 to 2012.

**Figure 6 entropy-22-00390-f006:**
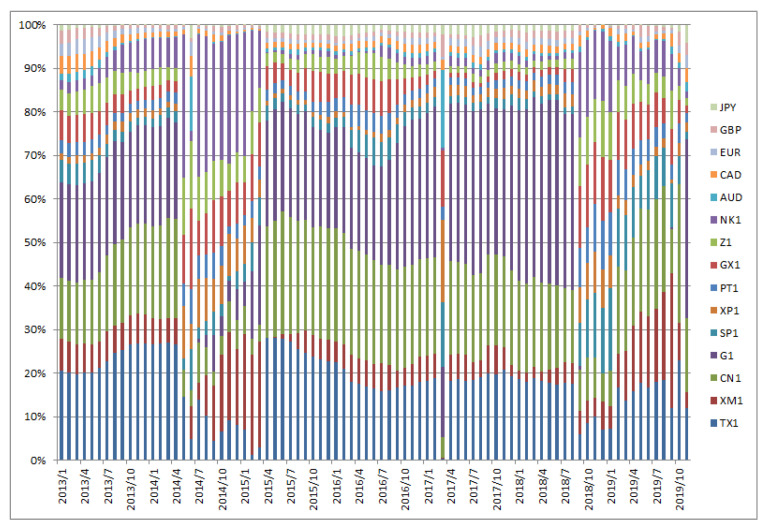
The allocation of the assets in the quaternion valued risk diversification (QVRD) portfolio construction from 2013 to 2019.

**Figure 7 entropy-22-00390-f007:**
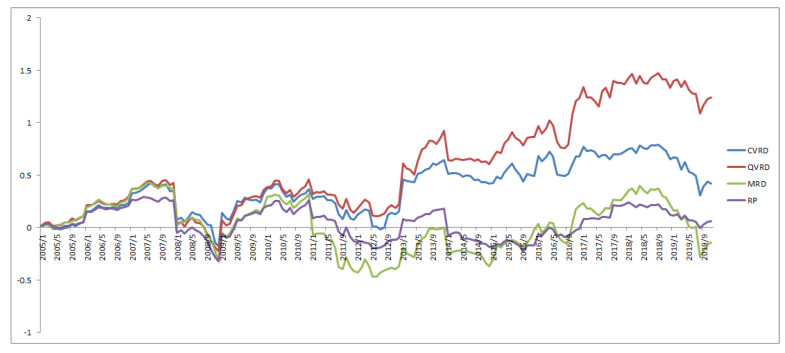
The cumulative returns of the RP and the MRD and the CVRD and the QVRD portfolio construction.

**Figure 8 entropy-22-00390-f008:**
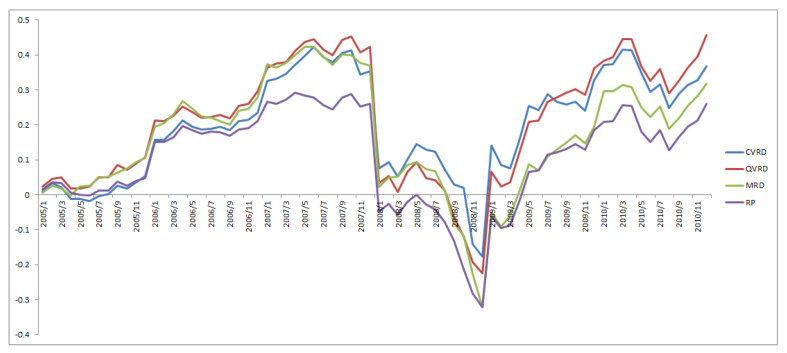
The cumulative returns of the RP and the MRD and the CVRD and the QVRD portfolio construction from 2005 to 2010.

**Figure 9 entropy-22-00390-f009:**
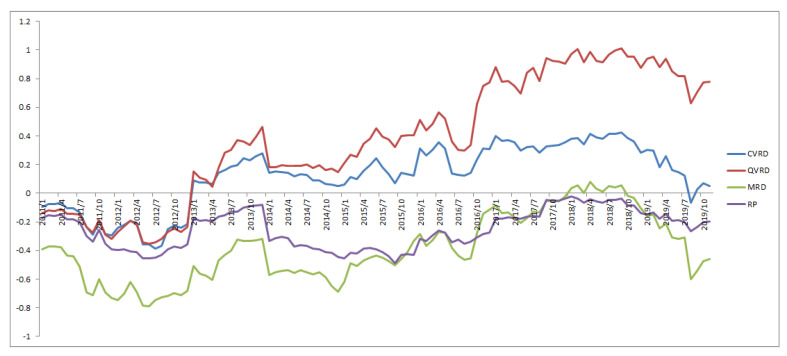
The cumulative returns of the RP and the MRD and the CVRD and the QVRD portfolio construction from 2011 to 2019.

**Table 1 entropy-22-00390-t001:** Asset list.

Name	Type	Definition
TX1	Commodity	10 Year T-Note Futures
XM1	Commodity	Australian 10 Year Bond
CN1	Commodity	Canadian Government 10 Year Note
G1	Commodity	Gilt UK
SP1	Index	S&P500
XP1	Index	S&P/ASX 200(Austraria)
PT1	Index	S&P/TSX 60 Index(Canada)
GX1	Index	DAX(German)
Z1	Index	FTSE100(UK)
NK1	Index	Nikkei225(Japan)
AUD	Currency	AUD/USD
CAD	Currency	CAD/USD
EUR	Currency	EUR/USD
GBP	Currency	GBP/USD
JPY	Currency	JPY/USD

**Table 2 entropy-22-00390-t002:** Descriptive statistics of the dataset.

	Mean	Standard Deviation	Skewness	Kurtosis
TX1	−0.00073	0.0209	0.0412	3.0058
XM1	−0.0293	0.0166	0.3360	5.6617
CN1	−0.00796	0.01996	0.228	2.9235
G1	−0.01266	0.0271	0.6344	10.6518
SP1	0.031	0.0114	−0.1345	12.6226
XP1	0.0187	0.0102	−0.35485	5.1535
PT1	0.0235	0.0107	−0.4535	13.110
GX1	0.03747	0.01287	0.12318	7.36026
Z1	0.01708	0.01096	0.01678	9.2329
NK1	0.02847	0.01424	−0.31261	9.06316
AUD	−0.0005	0.0079	−0.57578	13.3238
CAD	−0.00085	0.0058	0.2659	3.2109
EUR	−0.0039	0.00567	0.2659	3.2109
GBP	−0.0085	0.00588	−0.4954	11.6138
JPY	0.0003	0.00613	0.36956	4.8302

**Table 3 entropy-22-00390-t003:** Performance measures.

	RP	MRD	CVRD	QVRD
Return	0.061	−0.141	0.420	1.232
Standard Deviation	0.053	0.073	0.064	0.078
Sharp Ratio	0.022	−0.037	0.126	0.331
Omega Ratio	0.925	0.816	1.015	1.016
FT Ratio	0.388	0.385	0.496	0.512

**Table 4 entropy-22-00390-t004:** Averaged weights of the portfolios.

	RP(%)	MRD(%)	CVRD(%)	QVRD(%)
TX1	6.9	11.5	12.1	13.4
XM1	6.9	9.5	8.9	6.1
CN1	5.6	9.6	9.3	12.3
G1	6.7	12.1	10.1	13.8
SP1	6.4	8.4	7.9	7.1
XP1	6.5	7.5	7.7	5.0
PT1	6.4	7.7	5.5	5.9
GX1	6.6	7.0	5.9	8.9
Z1	6.3	6.2	5.6	7.9
NK1	6.5	4.3	6.9	7.1
AUD	7.1	4.1	5.6	3.3
CAD	6.7	4.4	4.5	3.5
EUR	6.6	4.4	4.2	2.0
GBP	7.1	4.9	3.9	2.3
JPY	7.1	5.1	4.1	1.5

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
