# Peer review of "Quaternion Valued Risk Diversification"

_entropy, 2020, doi:10.3390/e22040390_

Round 1

Reviewer 1 Report

Dear Authors

I read your manuscript "Quaternion Valued Risk Diversification" and will comment on it from a Finance perspective.

Firstly, sections of the manuscript need to be proofread or even rewritten. There are many run-on sentences and grammatically incorrect subclauses. I advise to use a professional proofreading service in order to make the manuscript better readable for the readers.

Secondly, the introduction is quite short. Some portfolio construction methods are introduced but not mentioned at all later on. I suggest to rewrite the introduction to have a clearer overview on why you introduce the QVRD. You mention that it is an extension to the CVRD but only compare the two. From a readers perspective, it would be more interesting to also have the standard and basic portfolio construction methods compared.

Thirdly, the portfolio construction and rebalancing is insufficiently described. It should be written in more detail on how the portfolio weights are calculated. Naturally, I assume that 256 trading days are used to calculate weights for the rebalancing at the end of the month and performance of the following month is then measured and recorded. Either way, it should be written out on how the authors carry this out.

Fourthly, the manuscript only shows that the QVRD outperforms the CVRD on the total observation window starting 2005. A more refined performance analysis over differing windows should be performed. Also, it is expected that those two construction methods outperform the classic construction methods. For the story and the message of the paper, it would be beneficial to demonstrate how the performance compares to classical construction methods which were mentioned in the introduction.

Lastly, the paper stops at describing that the QVRD outperforms the CVRD. As a reader, I wonder why that is. As a reader, I would also like to have a clear presentation of implications of these findings, which are missing at the moment.

As a minor comment, Figure 1 and Figure 2 are barely readable. I advise to either increase the size of the plot itself and the bars or to think about a different way of presenting the portfolio allocation. In particular, Figure 1 cannot be read.

Other than those comments, the mathematical foundation of the paper is solid and clean and does not need any revision.

Reviewer 2 Report

Report

The authors construct several portfolio optimization algorithms to minimize portfolio risk under certain constraints and propose the risk diversification portfolio construction method that incorporates quaternion risk. They show that the proposed method outperforms the conventional complex risk diversification portfolio.

I find their results interesting. I have the following comments to the authors to improve their paper:

  • It is well known that the estimation of the traditional portfolio selection model seriously overestimates its theoretic optimal return while Bai, et al. (2009) develop a bootstrap-corrected estimator to correct the overestimation. Do their approaches seriously overestimate/underestimate its theoretic optimal return/risk? Should they compare their results with those from others, like Bai, et al. (2009)?

  • There are some portfolio optimization algorithms that consider investors’ preferences in the algorithms, e.g. Li, et al. (2018). Could the authors consider investors’ preferences in their algorithms? Should they compare their results with those from others that consider investors’ preferences in the algorithms, like Bai, et al. (2009)?

  • The authors use Sharpe and Omega Ratios in the comparison. It is well known that using Sharpe Ratio has many problems and it is well known that using Omega Ratios direct has many problems also, see, for example, Guo, et al. (2017) and there are other risk measures that could be good also, see, for example, Guo, et al. (2019). Should the authors use the approaches of comparing Omega Ratios proposed by Guo, et al. (2017) in their analysis?

References

  1. Bai, Z.D., Liu, H.X., Wong, W.K., 2009. Enhancement of the applicability of Markowitz’s portfolio optimization by utilizing random matrix theory. Mathematical Finance, 19(4), 639-667.

  1. Guo, X., Jiang, X.J., Wong, W.K. (2017), Stochastic Dominance and Omega Ratio: Measures to Examine Market Efficiency, Arbitrage Opportunity, and Anomaly, Economies 5(4),

  1. Guo, X., Niu, C.Z., Wong, W.K. 2019, Farinelli and Tibiletti ratio and Stochastic Dominance, Risk Management, https://doi.org/10.1057/s41283-019-00050-2.

Round 2

Reviewer 1 Report

All comments have been adequately addressed. I suggest an acceptance.